



# Wind Field Reconstruction from Nacelle-Mounted Lidars Short Range Measurements

Antoine Borraccino[1], David Schlipf[2], Florian Haizmann[2], and Rozenn Wagner[1]

[1]DTU Wind Energy, Roskilde, Denmark.
[2]Stuttgart Wind Energy, University of Stuttgart, Germany.

*Correspondence to:* Antoine Borraccino (borr@dtu.dk)

**Abstract.** Profiling nacelle lidars probe the wind at several heights and several distances upstream the rotor. Yet, it is still unclear how to condense the lidar raw measurements into useful wind field characteristics such as speed, direction, vertical and longitudinal gradients (wind shear). In this paper, we demonstrate an innovative method to estimate wind field character-istics using nacelle lidar measurements taken within the induction zone. Model-fitting wind field reconstruction techniques are

applied to nacelle lidar measurements taken at multiple distances close to the rotor, where a wind model is combined with a simple induction model. The method allows robust determination of free stream wind characteristics. The method was applied to experimental data obtained with two different types of nacelle lidar (5-beam Demonstrator and ZephIR Dual-Mode). The reconstructed wind speed was within $0.5\%$ of the wind speed measured with a mast top-mounted cup anemometer at 2.5 rotor diameters upstream of the turbine. The technique described in this paper overcomes measurement range limitations of the

currently available nacelle lidar technology.

## 1  Introduction

In this section, we introduce the measurement principles of Doppler wind lidars, their benefits in the context of power perfor-mance verification and the need for new Wind Field Reconstruction (WFR) methods.

### 1.1  Why using nacelle lidars in power performance testing?

Nacelle-mounted two-beam lidars show promising capabilities to assess power performance (Wagner et al., 2014). Their use obviates the need to erect tall, costly and environmentally invasive meteorology masts, especially offshore. Investigating how to accurately estimate wind characteristics and quantify measurement uncertainties from such instruments is essential in order to consider using nacelle wind lidars in future standards for power performance testing.

The standards (IEC, 2016) require the measurement of hub height wind speed in order to measure a turbine's power curve.

This is typically achieved by mounting cup anemometers on a mast. The recommended distance from the turbine to the mast is 2.5 rotor diameters ($D_{rot}$). At this distance, the measured wind speed is considered a sufficient approximation of free stream wind speed. For testing a turbine's performance using nacelle lidars, measurements are commonly taken at the same distance. However, for large wind turbines ($D_{rot} \gtrsim 150\,\mathrm{m}$), the currently available nacelle lidar technology features insufficient range



capabilities of $300\,\mathrm{m}$–$400\,\mathrm{m}$. Additionally, the wind experiences at $2.5D_{\mathrm{rot}}$ a speed deficit up to $0.7\,\%$ due to the turbine's induction and thus is not in the 'free stream' (also true when the turbine is closely aligned with the mast direction). Consequently, a reliable method to estimate free stream wind characteristics from nacelle lidar short-range measurements is necessary.

## 1.2 Wind measurements with Doppler lidars

Doppler Wind Lidars (DWL) do not measure directly wind characteristics (Hardesty and Weber, 1987). They primarily sense backscattered light from particles moving with the wind. The return light originates from scatterers contained in a so-called probe volume located along the lidar beam. Wind Field Characteristics (WFC) – such as wind speed at hub height or vertical shear – are estimated combining velocity measurements taken over different Lines-Of-Sight (LOS). Except in the cases of co-located synchronised measurements (e.g. WindScanner (Vasiljevic, 2014) or multi-static systems), the different LOS velocities

result from probing the wind in several locations, therefore assumptions on the wind flow must be made.

A wind lidar is usually provided with or without embedded recontruction algorithms. In the first case, the lidar manufacturer implements its own methods to estimate WFC. Using embedded reconstruction algorithms, the lidar may be seen as a 'black box'. In the second case, the user himself condenses raw lidar data into useful information.

We chose the second approach to ensure transparency and flexibility. The model-fitting technique (Schlipf et al., 2012), initially developed for nacelle lidar systems to assist wind turbine control, was adapted to other nacelle lidar systems and applications (Schlipf, 2016). In this method, the LOS velocity ($V_{\mathrm{los}}$) and beam positions measurement data can be used to reconstruct wind characteristics using a model-fitting approach, where a wind model is defined by assuming e.g. horizontal homogeneity, vertical shear profiles, two- or three-dimensional wind vectors, etc. Knowing the DWL beam's location, one can

simulate $V_{\mathrm{los}}$ by projecting the modelled wind vector onto the LOS. A least squares problem is formulated: the reconstruction algorithm minimises the error between lidar-measured and model-estimated $V_{\mathrm{los}}$. As a result, the model WFC are obtained.

Commercial nacelle lidar systems may employ alternative methodologies, for example:

– the '*4-beam Wind Iris*' developed by *Avent Lidar Technology* assumes horizontal homogeneity at two different heights independently which yields simple analytical expressions to estimate horizontal wind speed and relative direction (Ma-

zoyer P., 2016). Wind shear and veer profiles are then calculated by the lidar in realtime, and hub height wind speed and direction are interpolated rather than directly measured. The former '*Wind Iris*' (2-beam system) also assumed horizontal homogeneity to estimate wind speed and direction at the sensed height.

– in the '*Dual-Mode*' system developed by *ZephIR Lidar* several reconstruction algorithms are implemented. One of them fits the raw measurements based on assumptions of horizontal wind flow, wind yaw misalignment and power law vertical

shear. Another one employs pairs of beams to estimate wind speed and direction similarly as a two-beam lidar, but at several heights below and above the hub (Medley et al., 2014). This latter algorithm allows arbitrary vertical shear profiles to be measured, as well as estimations of wind veer and rotor equivalent wind speed.



### 1.3 Motivations and research questions

From $V_{los}$ and other raw lidar measurements, the WFR relies on hypotheses on the wind field. The (in-)adequacy of these hypotheses plays a crucial role in condensing lidar raw measurements into information useful for various atmospheric and wind energy sciences applications, and affects the quality of the estimated WFCs. Consequently, WFR techniques must be
carefully described, and the underlying flow hypotheses clearly stated.

In this study we investigated the following research questions:

1. Can free stream wind characteristics be estimated using lidar measurements in the near flow of the turbine's rotor?

2. How do those lidar-estimated wind field characteristics compare to measurements from mast-mounted instrumentation?

Section 2 describes the model-fitting wind field reconstruction approach. We considered one 'static' wind model and its under-
lying physical assumptions are provided. Further, a combined wind-induction model is proposed, based on a simple induction model, allowing the retrieval of free stream wind characteristics from $V_{los}$ measured close to the turbine's rotor. Section 3 details the Nørrekær Enge measurement campaign providing the real-world testing environment of the newly developed wind field reconstruction technique. In Section 4, results are presented through comparisons between lidar-estimated and reference mast measurements of wind speed. WFCs have been reconstructed at several distances and heights above ground level (agl.).
Finally, we discuss in Section 5 potential improvements to the WFR methods, and several questions related to their application to nacelle lidars for power performance testing.

## 2   Wind Field Reconstruction

In this section, we define the concept of 'Wind Field Reconstruction' and describe the so-called model-fitting WFR technique used in this study, starting with the description of the necessary inputs, coordinate systems, lidar model and minimisation
problem. Next, several wind model examples are presented as well as a combined wind-induction model.

### 2.1   Methodology

Wind Field Reconstruction is the process of combining data containing information on wind vectors (e.g. $V_{los}$) in multiple locations in order to retrieve wind field characteristics relevant to the application. WFC can for instance be wind speed, direction, horizontal and vertical gradients – called shear and veer respectively for speed and direction –, turbulence (intensity, length
scales, etc).

With a Doppler wind system (lidar, sodar, radar, etc), performing WFR necessitates hypotheses on the spatial and temporal variations of the wind field. The reconstruction hypotheses and the WFCs define a wind model. Whenever possible, flow assumptions should rely on physical laws governing atmospheric flows. Depending on the needs and applications, WFR techniques can employ two types of wind models:

– 'static' models: the time dependency of the wind field variations is disregarded, i.e. stationarity is assumed. Typically, time-averaged measurement data provide the inputs to the reconstruction algorithms. Spatial flow assumptions are made,





for example on the number of components of the wind vector (one, two or three), on horizontal homogeneity or on the vertical shear profile.

– 'dynamic' models: both the time and spatial variations are accounted for. Flow models may be based on Navier-Stokes equations or Taylor's frozen turbulence hypothesis.

Dynamic WFR is suitable for turbine control applications (Raach et al., 2014; Towers and Jones, 2016) or evaluation of turbulence. For power performance assessment, requiring estimation of 10-minute statistics of wind characteristics, static WFR is adequate. This paper thus focuses on static wind models. We additionally chose to use the model-fitting WFR technique, further detailed in the rest of this section.

The flow chart in Figure 1 describes the model-fitting WFR methodology. The inputs to the process are:

– the wind model: the flow assumptions define the wind vector dimension (2D or 3D) and the WFC;

– the WFC initial values: in order to initialise the fitting process. Initial values have no influence on the fitted WFC values (output) if the solution of the minimisation problem is unique, and can for example be all set to 0;

– the lidar model: measurement trajectory and range configuration, point-like or volume-averaged $V_\mathrm{los}$ quantities;

– lidar raw measurement data: 10-minute average $V_\mathrm{los}$ and inclination angles (tilt and roll);

– for lidar systems with large motions (e.g. installed on floating wind turbines or platforms), additional sensors may be helpful (Schlipf et al., 2015).

In step (1), the lidar measurements are fitted to the wind model via an iterative optimisation process. At each iteration, the error between lidar-measured and model-simulated $V_\mathrm{los}$ is calculated. The fitting process minimises the $V_\mathrm{los}$ error and outputs

the fitted WFC values. In step (2), the wind field is estimated at the locations of interest applying the wind model to the WFC values thus yielding reconstructed wind parameters – for example horizontal wind speed at $2.5 D_\mathrm{rot}$ upstream and hub height.

## 2.2   Formulation and solving of the minimisation problem

In order to fit WFC to the lidar measurements (step (1) in Fig. 1), a least squares (LS) problem is formulated. The objective is to minimise the error between lidar-measured $\boldsymbol{V}_\mathrm{los}$ and model-simulated $\widehat{\boldsymbol{V}}_\mathrm{los}$ where the error is defined as

$$\left\| \boldsymbol{V}_\mathrm{los} - \widehat{\boldsymbol{V}}_\mathrm{los} \right\|_2 = \sqrt{\sum_{i=1}^{N_\mathrm{los}} \left( V_{\mathrm{los},i} - \widehat{V}_{\mathrm{los},i} \right)^2}. \tag{1}$$

Note that $\boldsymbol{V}_\mathrm{los}$ and $\widehat{\boldsymbol{V}}_\mathrm{los}$ are vectors of length corresponding to the number $N_\mathrm{los}$ of $V_\mathrm{los}$ measurements .

For linear wind models, such as when flow homogeneity is assumed, the solution of the LS problem can be obtained by matrix inversion (Schlipf et al., 2012). More complex flow models are usually non-linear (see Sect. 2.4). To solve a non-linear LS problem, optimisation algorithms may be utilised. In this paper, we selected the Levenberg-Marquardt algorithm

(Marquardt, 1963), also called the Damped Least-Squares – an optimal gradient-based minimisation method – to solve the non-linear LS problem (Eq. 1).

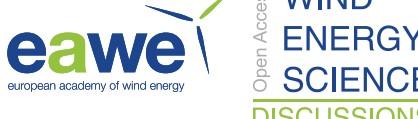



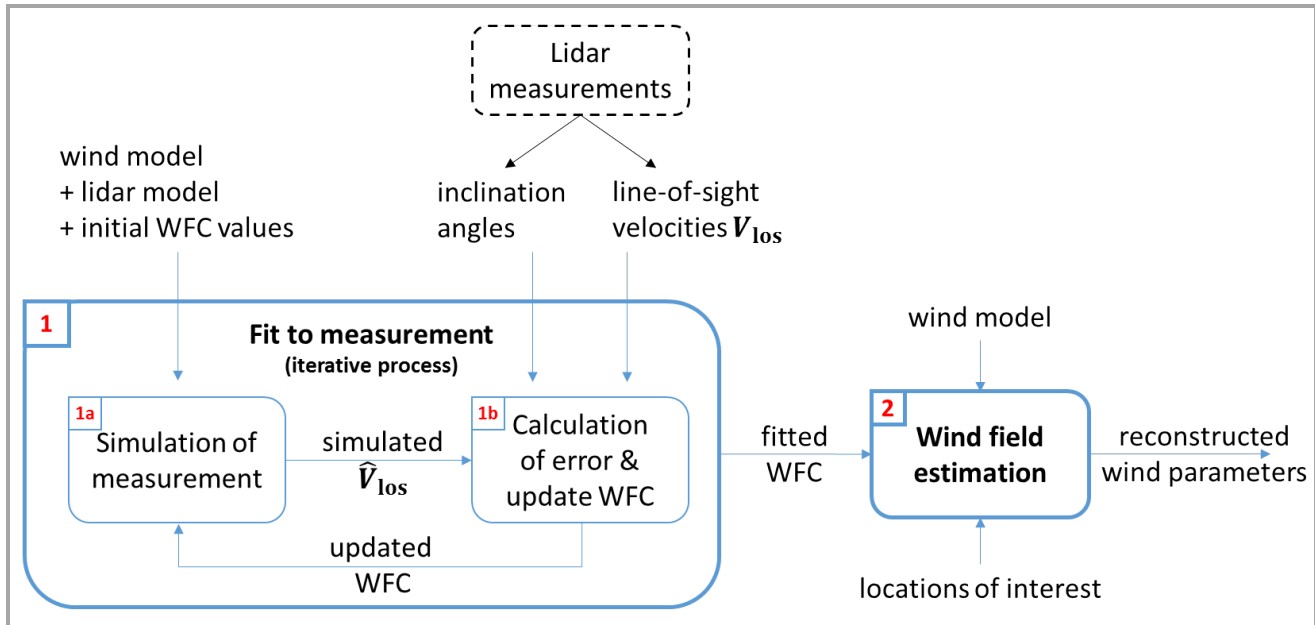

**Figure 1.** Flow chart of the model-fitting Wind Field Reconstruction methodology.

## 2.3 Lidar model

### 2.3.1 Coordinate systems

When performing WFR, the locations at which the lidar measures $V_{los}$ in relation to the lidar position on the nacelle play a crucial role in accurately simulating the measurements (see step (1a) in Fig. 1). Coordinate systems (CS) must therefore be carefully defined. Moreover, the mathematical definition of the wind model may be simpler in one CS or another. Adequately selecting the CS allows an easier and more robust fitting of the lidar measurements to the wind model.

The developed WFR method employs several CS. We here define the lidar, hub and wind CS (Fig. 2).

The **lidar CS** $(\mathcal{L}, \overrightarrow{x}_{\mathcal{L}}, \overrightarrow{y}_{\mathcal{L}}, \overrightarrow{z}_{\mathcal{L}})$ is a right-handed Cartesian orthonormal system with its origin at the point where the lidar emits its beam and the X-axis defined by the lidar optical centerline, pointing upwind for the power curve application. The location of measurement point $j$ in the lidar CS is denoted $(x_{j,\mathcal{L}}, y_{j,\mathcal{L}}, z_{j,\mathcal{L}})$ and derived directly from the measurement ranges and lidar geometry (e.g. opening angles).

The **hub CS** $(\mathcal{H}, \overrightarrow{x}_{\mathcal{H}}, \overrightarrow{y}_{\mathcal{H}}, \overrightarrow{z}_{\mathcal{H}})$ origin is at the center of the rotor plane. The hub CS is obtained by transforming the lidar CS with two rotations (tilt around $\overrightarrow{y}_{\mathcal{L}}$ and roll around $\overrightarrow{x}_{\mathcal{L}}$) and one translation corresponding to the lidar position in the hub CS $(x_{L,\mathcal{H}}, y_{L,\mathcal{H}}, z_{L,\mathcal{H}})$. The X-axis of the hub CS points downwind. $(\overrightarrow{x}_{\mathcal{H}}, \overrightarrow{y}_{\mathcal{H}})$ defines a horizontal plane.

The **wind CS** $(\mathcal{H}, \overrightarrow{x}_{\mathcal{W}}, \overrightarrow{y}_{\mathcal{W}}, \overrightarrow{z}_{\mathcal{W}})$ shares the origin of the hub CS (no translation) and its X-axis is aligned with the mean wind vector – in other words it is obtained by using the fitted relative wind direction and vertical flow angle and applying two rotations.

Since the lidar follows the turbine's motion, the three aforementioned CS are independent of the turbine's yaw position.





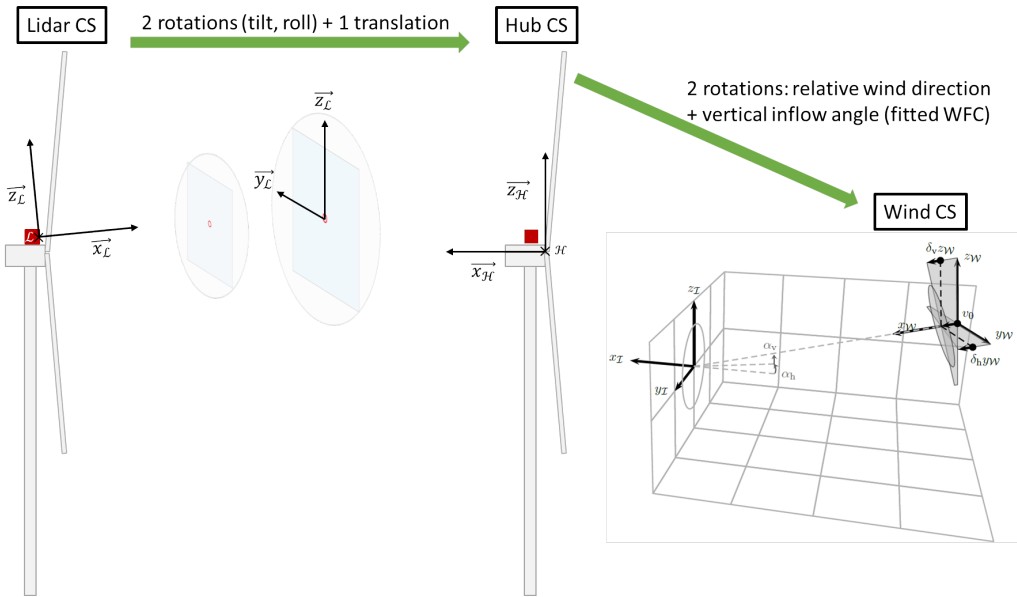

**Figure 2.** Schematic and relation between used lidar, hub and wind coordinate systems.

### 2.3.2 Measurement simulation

Simulating the lidar consists in computing the LOS velocities. To do so, the beam coordinates are expressed in the same CS as the one used for defining the wind model. First, in the lidar CS, the coordinates $(x_{j,\mathcal{L}}, y_{j,\mathcal{L}}, z_{j,\mathcal{L}})$ of measurement point $j$ are directly derived from the lidar trajectory and measurement range. The trajectory may be defined for example by the lidar cone or half-opening angles. Then, in the hub CS, the normalised vector $\overrightarrow{n}_{j,\mathcal{H}}$ towards measurement point $j$ is given by:

$$\overrightarrow{n}_{j,\mathcal{H}} = \begin{bmatrix} x_{n,j,\mathcal{H}} \\ y_{n,j,\mathcal{H}} \\ z_{n,j,\mathcal{H}} \end{bmatrix} = \frac{1}{\sqrt{(x_{j,\mathcal{L}}^2 + y_{j,\mathcal{L}}^2 + z_{j,\mathcal{L}}^2)}} \left( \begin{bmatrix} x_{L,\mathcal{H}} \\ y_{L,\mathcal{H}} \\ z_{L,\mathcal{H}} \end{bmatrix} - \begin{bmatrix} x_{j,\mathcal{H}} \\ y_{j,\mathcal{H}} \\ z_{j,\mathcal{H}} \end{bmatrix} \right). \tag{2}$$

The lidar LOS velocities can be modelled either as point-like or volume-averaged quantities. If the volume-averaged lidar model is used, the simulation of the measurements requires the integration of the probe volume weighting function (Sathe et al., 2011; Angelou et al., 2012). For static WFR, the difference between volume-averaged and point-like LOS velocities is only significant if the mean wind field along the beam path features large non-linearities. For the sake of simplicity, we only considered the point-like model. Hence, the simulation of the lidar measurements is the projection of the local wind vector $\overrightarrow{U}_j$ onto LOS$_j$, mathematically obtained by

$$\widehat{V}_{\text{los},j} = \overrightarrow{n}_{j,\mathcal{H}} \cdot \overrightarrow{U}_j, \tag{3}$$

where $\cdot$ is the scalar product.





## 2.4 Wind and induction models

In this paragraph, we propose, describe and mathematically define the two static flow models employed in this analysis:

1. a wind model assuming horizontal flow, vertical shear and veer profiles;

2. the previous wind model combined to a simple induction model.

5 Let $[u, v, w]$ be the three components of the wind vector $\overrightarrow{U}$. A static wind model is defined by the function $f$ as follows:

$$\overrightarrow{U}(x, y, z) = f(x, y, z, p_1, \ldots, p_N), \tag{4}$$

where $x, y, z$ are the field coordinates in an arbitrary CS, and $p_1, \ldots, p_N$ are the WFC. In the general case, the wind vector is three-dimensional (3D). In flat terrain or offshore, the vertical component $w$ of the wind vector can reasonably be neglected. The flow is assumed horizontal, and thus the wind vector is two-dimensional (2D): $\overrightarrow{U} = [u, v, 0]$.

### 2.4.1 Wind model

The wind model hypotheses are: horizontal homogeneity; wind speed varying with height according to a chosen shear profile; homogeneous relative wind direction (no veer). $V_0$ denoting the horizontal wind speed at hub height $H_{\text{hub}}$, $\theta_r$ the relative direction, and $p_{\text{shear}}$ a shear parameter, the wind model in the hub CS is given by

$$\overrightarrow{U}(x_{\mathcal{H}}, y_{\mathcal{H}}, z_{\mathcal{H}}) = \overrightarrow{U}(z_{\mathcal{H}}) = f(z_{\mathcal{H}}, V_0, \theta_r, p_{\text{shear}}). \tag{5}$$

15 Vertical shear profiles depend mainly on atmospheric stability, terrain elevation and roughness. Examples of shear profiles are:

(i) logarithmic-law:

$$V(z_{\mathcal{H}}) = \frac{v_\star}{\kappa} \log\left(\frac{z_{\mathcal{H}} + H_{\text{hub}}}{z_0}\right), \tag{6}$$

where $v_\star$ is the friction velocity, $\kappa$ the Von Kármán constant, $z_0$ the roughness length.

(ii) power-law:

$$V(z_{\mathcal{H}}) = V_0 \left(\frac{z_{\mathcal{H}} + H_{\text{hub}}}{H_{\text{hub}}}\right)^\alpha, \tag{7}$$

where $\alpha$ is the shear exponent.

(iii) linear:

$$V(z_{\mathcal{H}}) = V_0 \delta_v z_{\mathcal{H}}, \tag{8}$$

where $\delta_v$ is the linear vertical shear gradient.



The logarithmic-law is only valid for neutral stratification. For buoyancy driven (BD) wind profiles, Högström (1988) proposed empirical formulas to account for stability and correct the log-law. BD profile corrections were fitted to measurements and are not valid for very stable stratification. The determination of stability classes – usually based on the Obukhov length or on the bulk Richardson number – is sensitive to the employed methods (Holtslag et al., 2014). Moreover, a DWL cannot determine
the stability class on its own and external instruments would be required. Therefore, the logarithmic profile was not considered in this study.

In wind engineering applications, the power-law is often used. When measurements are taken in a narrow heights range – such as nacelle lidars measuring within the rotor area – the power-law is a simple and accurate enough approximation of the wind profile if no further information is available. In addition, the IEC (2016) norm suggests its use to characterise shear
profiles.

The linear profile is an even simpler approximation of the wind profile. The largest non-linearities in the log- or power-law profiles are located close to the ground. In situations where measurements are taken at heights sufficiently far from the ground – we propose above $\approx 30\,\mathrm{m}$ as a rule of thumb – a linear profile may be considered. Contrary to the log- and power-laws requiring knowledge of the measurement height agl., no site-specific information is necessary. However, the linear profile does
not physically characterise wind profiles. Further, we only consider the power-law wind profile.

### 2.4.2   Combined wind-induction model

By harnessing energy from the wind, an operating turbine creates an induction zone upstream its rotor (Sørensen, 2016; Simley et al., 2016): the closer to the turbine, the lower the wind speed. Adequately modelling wind speed variations in the induction zone constitutes the main challenge of the WFR for nacelle lidar measurements taken e.g. within 0.5 to 2 rotor diameters
upstream the turbine. Computational Fluid Dynamics simulations (Troldborg and Meyer Forsting, 2016) have shown that, at upstream distances larger than $0.5D_{\mathrm{rot}}$, the induction becomes insensitive to the blades' aero-elastic properties or to the turbine's control strategy. Except in the direct proximity of the rotor plane, the induction zone of a wind turbine is self-similar (see Sect. 5.1 for discussion). The 'intensity' of the induction however depends on the thrust generation capabilities of the turbine, which may be quantified via an induction factor.
The vortex cylinder model applied to the actuator disk concept yields a simple expression characterising the induction (Branlard and Gaunaa, 2015; Medici et al., 2011) that can be integrated into a WFR model. This simple induction model is one-dimensional. It only is a function of the streamwise distance to the turbine. If both vertical shear (with a power law profile) and induction effects are accounted for, the combined wind-induction model takes the following form:

$$\overrightarrow{U}\left(x_{\mathcal{H}}, y_{\mathcal{H}}, z_{\mathcal{H}}\right) = \overrightarrow{U}\left(x_{\mathcal{H}}, z_{\mathcal{H}}\right) = f\left(x_{\mathcal{H}}, z_{\mathcal{H}}, V_0, \theta_r, \alpha, a\right), \tag{9}$$

where $a$ is the induction factor. With $\overrightarrow{U} = [u, v, 0]$ defined in the hub CS, the cross-stream wind component negligibly contributes to the generation of thrust by the turbine. The analytical induction function thus applies to the streamwise component





of the wind vector and is given by:

$$\frac{u\left(x_{\mathcal{H}}, z_{\mathcal{H}}=0\right)}{u_{\infty}}=1-a\left[1+\frac{\xi}{\sqrt{1+\xi^{2}}}\right], \tag{10}$$

where $u_{\infty}$ is the streamwise component of the free stream wind speed $V_{\infty}$ at hub height, $\xi = x_{\mathcal{H}}/R_{\mathrm{rot}}$ is the longitudinal coordinate in the hub CS adimensionned by the rotor radius $R_{\mathrm{rot}}$. Combining Eq. 10 with the power-law shear profile in Eq. 7, the wind-induction model is given by

$$V\left(x_{\mathcal{H}}, z_{\mathcal{H}}\right)=\sqrt{u^{2}\left(x_{\mathcal{H}}, z_{\mathcal{H}}\right)+v^{2}\left(z_{\mathcal{H}}\right)}=\left(\frac{z_{\mathcal{H}}+H_{\mathrm{hub}}}{H_{\mathrm{hub}}}\right)^{\alpha}\sqrt{u_{\infty}^{2}\left(1-a\left[1+\frac{\xi}{\sqrt{1+\xi^{2}}}\right]\right)^{2}+v_{\infty}^{2}}, \tag{11}$$

with $u_{\infty}=V_{\infty}cos(\theta_{r})$ and $v_{\infty}=V_{\infty}sin(\theta_{r})$.

The wind-induction model yields four WFC: the hub height free stream wind speed $V_{\infty}$ and relative direction $\theta_{r}$, the shear exponent $\alpha$ and the induction factor $a$.

## 3 Testing Environment: The Nørrekær Enge Measurement Campaign

The Unified Turbine Testing (UniTTe) research project aims at establishing turbine performance testing procedures applicable in any type of terrain, i.e. for onshore simple or complex sites as well as offshore. Within UniTTe, a 7-month measurement campaign was conducted in Nørrekær Enge (NKE), between June 2015 and January 2016.

This section provides details on the site, wind farm layout, mast instrumentation and nacelle lidars setup.

### 3.1 Terrain, climate and wind farm

The NKE wind farm is located in northern Jutland, Denmark and owned by Vattenfall[1]. The park comprises one row of 13 Siemens $2.3\,\mathrm{MW}$ turbines having a rotor diameter $D_{\mathrm{rot}}$ of $93\,\mathrm{m}$ and a hub height $H_{\mathrm{hub}}$ of $80\,\mathrm{m}$ agl. The orientation of the turbines' row is $75°-255°$. The site is mainly characterised by open crop fields and the terrain flatness. In the vicinity of turbine number 4 (T04), except for the turbines' foundations, variations in elevation of $\pm1\,\mathrm{m}$ are observed (see Fig. 3). The prevailing wind direction is West. In Jutland, such western winds often feature high speeds (Peña et al., 2016).

### 3.2 Meteorological mast and turbine instrumentation

A meteorological mast was installed $232\,\mathrm{m}$ from T04 approximately in the $103°$ direction (see Fig. 3). The mast instrumentation complies with the requirements of the standards for power performance measurement (IEC61400-12-1, 2005):

- one top-mounted cup anemometer at $80\,\mathrm{m}$ agl.;

- three cup anemometers and wind vanes at $33.5\,\mathrm{m}$, $57.5\,\mathrm{m}$ and $78\,\mathrm{m}$ agl.;

---

[1]Find more information on: https://corporate.vattenfall.dk/




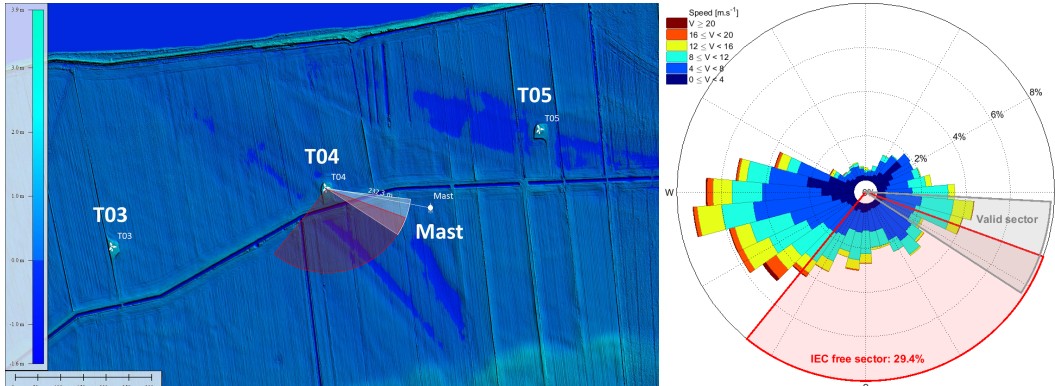

**Figure 3. Left:** elevation map in the vicinity of T04, Nørrekær Enge wind farm. Map: DHM/Terrain (0.4 m grid). Source: *Styrelsen for Dataforsyning og Effektivisering.* **Right:** wind rose during the NKE campaign, measured by the top-mounted cup anemometer and wind vane.

– one sonic anemometer at 76 m agl.;

– other sensors: air temperature at 2 m and 78 m, relative humidity at 78 m, atmospheric pressure at 77 m, and precipitation at 20 m agl.

More details about the measurement system of the NKE experiment can be found in Vignaroli and Kock (2016).

### 3.3 Nacelle-mounted lidars: measurement characteristics and configuration

Two commercially developed profiling nacelle lidar systems were mounted on the nacelle of T04 (see Fig. 4): a 5-beam Avent Demonstrator (5B-Demo) and a ZephIR Dual-Mode (ZDM).

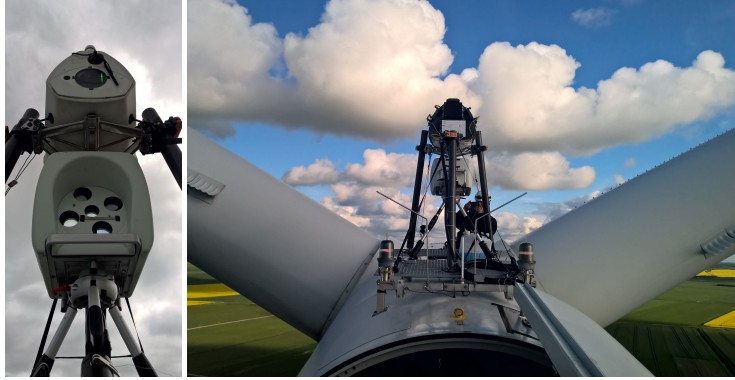

**Figure 4.** Measurement campaign in Nørrekær Enge (Denmark): the 5-beam Avent Demonstrator (bottom) and ZephIR Dual-Mode (top) lidars mounted on a Siemens 2.3 MW turbine.

The 5-beam Demonstrator is, likewise the other *Avent* lidars, a pulsed system measuring $V_{los}$ at several distances simultaneously along each LOS by range-gating. The five LOS form a square trajectory (4 corners and center). They are measured successively at 1 Hz, thus a complete cycle takes 5 s. Being a pulsed system, the turbine's blades are in the lidar's 'blind zone' and cannot contaminate $V_{los}$; the only effect of blade blockage is a reduced LOS availability.



The ZephIR Dual-Mode is a continuous wave (CW) system featuring a variable focus to interrogate multiple distances successively. Each distance is conically scanned. ZDM samples $V_{los}$ at high frequency ($\approx 50\,\text{Hz}$). For each $V_{los}$ measurement, the azimuthal position is recorded as the center of the probed circle arc. We developed an algorithm averaging raw high frequency $V_{los}$ measured in separate azimuthal sectors and yielding a pseudo '48-beam lidar'. When a lidar beam hits a blade, $V_{los}$ can

be significantly contaminated by the presence of the blade. Consequently, the data was quality controlled using recordings of Doppler spectra in order to remove invalid $V_{los}$ measurements such as in the event of full or partial blade blockage and low Doppler signals (due to e.g. moving grass close to the ground).

Prior to the NKE campaign, the 5B-Demo and ZDM lidars were calibrated at DTU's test section for large wind turbines,

Høvsøre, Denmark. The calibration ensures traceability of the lidar measurements to international systems of units and provide estimates of the $V_{los}$ measurement uncertainty. The calibration methodology employed the so-called 'white-box' approach (Borraccino et al., 2016). Calibration reports and more details on the lidars' measurement principles can be found in Borraccino and Courtney (2016a) and Borraccino and Courtney (2016b).

The lidars were aligned with T04's axis via their internal alignment systems (visible laser lights) and measured to $\lesssim 0.5°$.

Their position ($x_{L,\mathcal{H}}$, $y_{L,\mathcal{H}}$, $z_{L,\mathcal{H}}$) in the hub coordinate system (see Sect. 2.3.1) was measured with a total station: for both lidars, the distance from the rotor plane is $x_{L,\mathcal{H}} \approx 2.5\,\text{m}$.

Table 1 provides the range configuration of the 5B-Demo and ZDM lidars in NKE, and the time spent at each distance during one cycle for ZDM. The corresponding measurement trajectories are visualised in Fig. 5.

**Table 1.** Configuration of lidars measurement distances during the Nørrekær Enge campaign.

| Lidar | Configured measurement distances [m] | | | | | | | | | | |
|---|---|---|---|---|---|---|---|---|---|---|---|
| 5B-Demo | - | - | 49 | 72 | 95 | 109 | 121 | 142 | 165 | 188 | 235 | 281 |
| ZDM | 10 (5 s) | 30 (10 s) | - | - | 95 (10 s) | - | 120 (10 s) | | | | 235 (15 s) | - |
| **Lidar** | **Distances in hub CS adimensionned by $D_{rot}$ [−]** | | | | | | | | | | |
| 5B-Demo | - | - | 0.5 | 0.75 | 0.99 | 1.14 | 1.27 | 1.5 | 1.75 | 1.99 | 2.5 | 2.99 |
| ZDM | 0.08 | 0.30 | - | - | 0.99 | - | 1.26 | - | - | - | 2.5 | - |





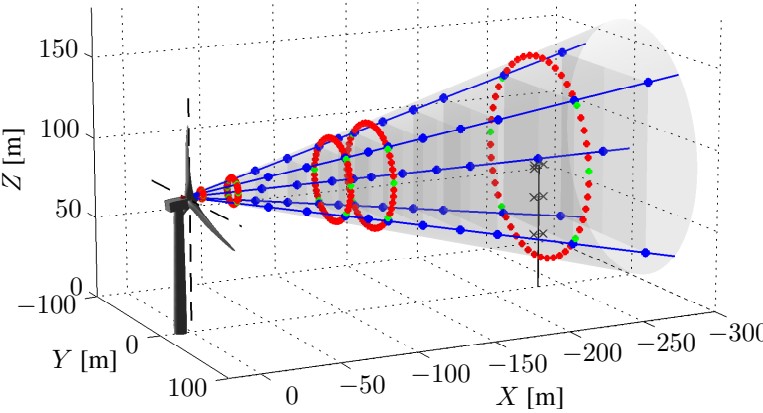

**Figure 5.** Lidar measurement trajectories in NKE. **In blue:** 5-beam Avent Demonstrator. **In red and green:** ZephIR Dual-Mode, the 48 azimuth-averaged LOS (red), including the 6 LOS (green) considered in the reconstruction cases of this paper.

### 3.4 Data filtering

Following the procedure for assessing free sectors in Annex A of (IEC, 2016), no significant obstacle exists in the vicinity of T04 in NKE other than neighbouring turbines causing wakes. The disturbed sectors were calculated for the meteorology mast and for the nacelle lidars using the IEC formula for wakes adapted to nacelle lidars (Wagner and Davoust, 2013). The resulting

5 undisturbed sectors are $110° - 219°$ and $318° - 22°$. Note that this procedure is conservative.

Practically, wake sectors were observed approximately for wind directions $\in [28°, 84°] \cup [240°, 300°]$, based on turbulence intensity measured by the mast top-mounted anemometer. Additionally, the error between the lidar-reconstructed (with the wind model from Sect. 2.4) and mast-measured wind speed is analysed prior to filtering (Fig. 6). Large errors due to wakes are observed in consistent sectors.

10 In order to compare the lidar and the mast measurements, we selected a sector of $[93°, 123°]$, close to the turbine-mast direction of $103°$.

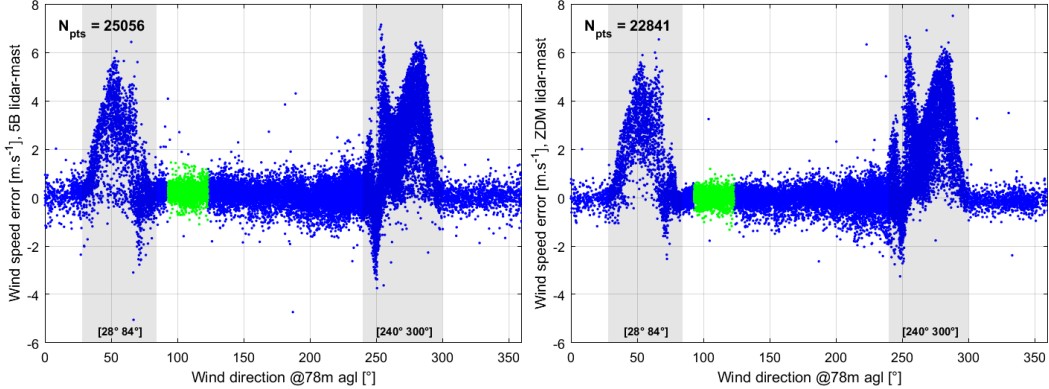

**Figure 6.** Analysis of horizontal wind speed error (lidar-reconstructed at $235\,\mathrm{m}$ minus mast-measured). Gray shaded areas show wind directions with wakes. Data in green are in the selected sector. **Left:** 5B-Demo lidar. **Right:** ZDM lidar.





Valid 10-minute measurement periods are obtained by filtering datasets as follows:

- **Mast**: wind direction measured by the wind vane at $78\,\mathrm{m}$ agl. $\in [93°, 123°]$;

- **Turbine**:
    - connected to the grid without disruption;
    - operating without disruption;
    - moderate yaw misalignment – also called relative wind direction. Only periods where the yaw misalignment measured by the spinner sonic anemometer $\in [-10°, 10°]$ are considered.

- **5B-Demo lidar**:
    - LOS availability $> 30\,\%$. Note that, due to the blades roots thickness, the bottom beams encounter blockage more frequently than the central or top beams. Thus, the LOS availability threshold is set to a relatively low value;
    - Carrier-to-Noise Ratio (CNR) $> -20\,\mathrm{dB}$ for all five LOS, using the realtime data. Additionally, the presence of the mast may bias the 10-minute average $V_{\mathrm{los}}$ towards 0 when one of the lidar's beam hits the mast frame. Such 10-minute periods were removed from the analysis by thresholding the difference between the observed maximum and mean CNR. The threshold was set to an arbitrary value of $15\,\mathrm{dB}$;
    - Tilt and roll measured by the lidar's internal inclinometers are real numbers;
    - All of the five LOS must pass the filtering for the period to be valid.

- **ZDM lidar**:
    - LOS availability $> 30\,\%$ independently of the azimuth sector considered. This criteria is more often met when the beam points upwards than downwards for the same reasons as for 5B-Demo;
    - LOS count $> 50$. The count is the number of times ZDM attempts to measure $V_{\mathrm{los}}$ in a 10-minute period. Combined with the $> 30\,\%$ minimum availability, this filter ensures a minimum quantity of data points to compute the average $V_{\mathrm{los}}$ in the considered azimuth sectors.
    - Tilt and roll measured by the lidar's internal inclinometers are real numbers;
    - Fog filtering: periods where fog is detected are rejected. Fog biases lidar measurements, particularly for a CW system. Abnormally strong backscatter returns from short ranges are observed. When the focus is at longer distances, these short-range returns are in the tail of the lidar Lorentzian weighting function. As a result, the measured $V_{\mathrm{los}}$ does not correspond to the LOS velocity at the expected measurement distance, due to the lidar volume-averaging properties. Fog events were detected by thresholding both the mean backscatter at $10\,\mathrm{m}$ and its ratio with the backscatter at the range of interest.
    - In the employed reconstruction case, 6 LOS are used (see green dots in Fig. 5). Each LOS must pass the filtering for the period to be valid.




## 4    Results

In this section, the results obtained with the WFR methods are presented through comparisons between the lidar-estimated (reconstructed) and mast-measured horizontal wind speeds.

The data analysis is performed on joint datasets. A valid period is consequently obtained when the mast, turbine, and
both lidars have successfully passed the filters detailed in Section 3.4. Joint datasets allow to compare the results of various reconstruction cases, as the variability of the wind conditions cannot be source of deviations. On the negative side, the amount of data points is significantly reduced.

### 4.1    Reconstruction with wind model

The flow is here assumed to be horizontal (no vertical component). The wind model assuming a power law shear profile (see
Sect. 2.4) is applied to the lidars $V_{\mathrm{los}}$ measurements taken:

   – for ZDM, at $235\,\mathrm{m}$. This corresponds to the mast-turbine distance ($2.5D_{\mathrm{rot}}$);

   – for 5B-Demo, at $188\,\mathrm{m}$. Due to operational issues during the campaign (dust and salts accumulating on the optical
      head's window), using the $235\,\mathrm{m}$ range requires stricter quality filtering leading to very small datasets (less than 200 data
      points). The considered $188\,\mathrm{m}$ distance ($2.0D_{\mathrm{rot}}$) is the shortest one accepted for power performance testing by the IEC
(2016) norm.

Figure 7 displays scatter plots of the horizontal wind speeds – denoted $V_{\mathrm{hor}}$ – measured by the top-mounted cup anemometer, and estimated at $80\,\mathrm{m}$ agl. from the lidars' measurements. Unforced (red) and forced (black) linear regressions results are also displayed. Compared to the mast measurements, both lidars overestimate the wind speed by $1-1.5\%$ (forced regression), with consistent coefficients of determination $R^2 > 0.993$.

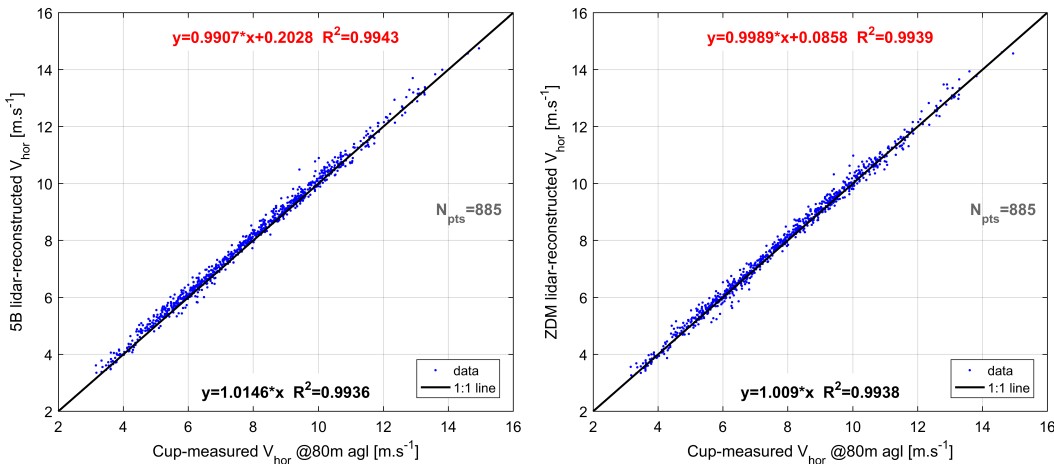

**Figure 7.** Comparison between mast-measured and lidar-estimated horizontal wind speed at $80\,\mathrm{m}$ height agl.

**Left**: 5B-Demo lidar, using 5 LOS, at $2.0D_{\mathrm{rot}}$. **Right**: ZDM lidar, using 6 LOS, at $2.5D_{\mathrm{rot}}$.



In the considered wind model, a shear exponent is fitted to the lidar measurements, thus allowing wind speed estimations at any desired height. The selected height should however remain within the probed lidar heights, approximately $40 - 120\,\mathrm{m}$ agl. in NKE. This is illustrated by Figure 8, where the wind speed is estimated by the lidar at $57.5\,\mathrm{m}$ agl. for comparison with a side-mounted cup anemometer. Although no $V_{\mathrm{los}}$ measurement is taken at this particular height, a high level of agreement

between the lidar estimates and mast measurements of $V_{\mathrm{hor}}$ is obtained, thus demonstrating a satisfactory level of the adequacy of the fitted shear profile.

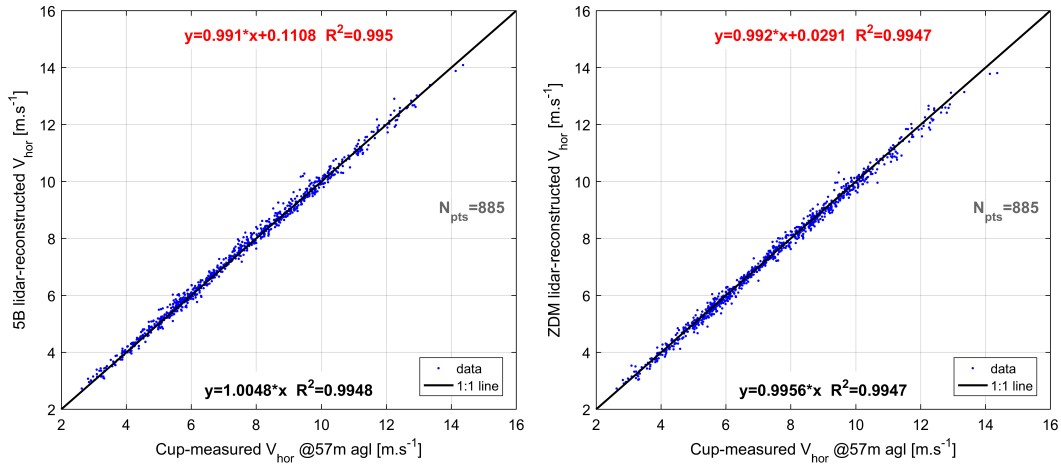

**Figure 8.** Comparison between mast-measured and lidar-estimated horizontal wind speed at $57.5\,\mathrm{m}$ height agl.
**Left**: 5B-Demo lidar, using 5 LOS, at $2.0D_{\mathrm{rot}}$. **Right**: ZDM lidar, using 6 LOS, at $2.5D_{\mathrm{rot}}$.

### 4.2    Reconstruction with combined wind-induction model

In this paragraph, the combined wind-induction model is used to perform WFR on the lidars' short-range measurements. Multiple distances sufficiently separated from one another are required to fit the simple induction model (Eq. 11).

Only $V_{\mathrm{los}}$ measurements taken close to the rotor, thus experiencing a significant wind speed deficit, were selected as inputs to the reconstruction. We chose the first four ranges, from $0.5$ to $1.15D_{\mathrm{rot}}$ for the 5B-Demo lidar, and the three distances from $0.3$ to $1.25D_{\mathrm{rot}}$ for ZDM. These distances are the closest to the turbine's rotor for which the induction may be considered self-similar. Hence, the $10\,\mathrm{m}$ range measured by ZDM was discarded. In addition, the wind field experiences larger longitudinal gradients close to the turbine. The fitting of the induction factor is thus facilitated and more robust. Finally, proving the concept

of WFR using lidar short-range measurements can only be achieved if the free stream measurements are discarded from the inputs of the WFR algorithms. Although the $V_{\mathrm{los}}$ measurements are taken close to the rotor, lidar estimates of wind speed can be reconstructed from the fitted WFC at any distance upstream and any height.

In Figure 9, $V_{\mathrm{hor}}$ is estimated at $2.5D_{\mathrm{rot}}$ upstream (i.e. $\xi = -5$, see Eq. 9) and hub height – by using the fitted free stream wind speed $V_\infty$, induction factor $a$, and shear exponent $\alpha$. The comparisons between the lidar-estimated and mast-measured

wind speed show an excellent level of agreement with gain errors of $+0.6\,\%$ and $-0.4\,\%$ for 5B-Demo and ZDM respectively.




The scatter is slightly reduced in comparison to Figure 7, with $R^2$ values $> 0.994$. Note also that the mast dataset is exactly the same in both Figures 7 and 9.

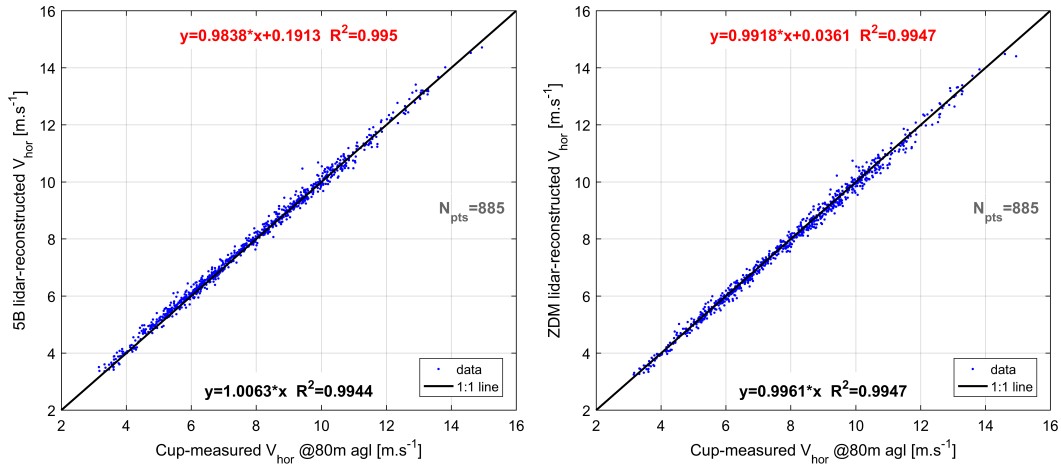

**Figure 9.** Comparison between mast-measured and lidar-estimated horizontal wind speed at hub height and $2.5D_{\mathrm{rot}}$ using short-range measurements. **Left**: 5B-Demo lidar, using 5 LOS and 4 ranges. **Right**: ZDM lidar, using 6 LOS and 3 ranges.

Wind speed comparisons at $57.5\,\mathrm{m}$ agl. are displayed in Figure 10. Although the fitted WFC are the same as in the hub height comparison (Fig. 9), the lidar estimates deviate from the cup measurements by $2\,\%$ for ZDM and $0.7\,\%$, for 5B-Demo. Using short-range $V_{\mathrm{los}}$ measurements, the lidar trajectories cover a narrower range of heights (in this case, $\sim 60 - 100\,\mathrm{m}$ agl.). The comparison height is outside this range, which may explain the larger deviations observed here in the lidar estimates of wind speed.

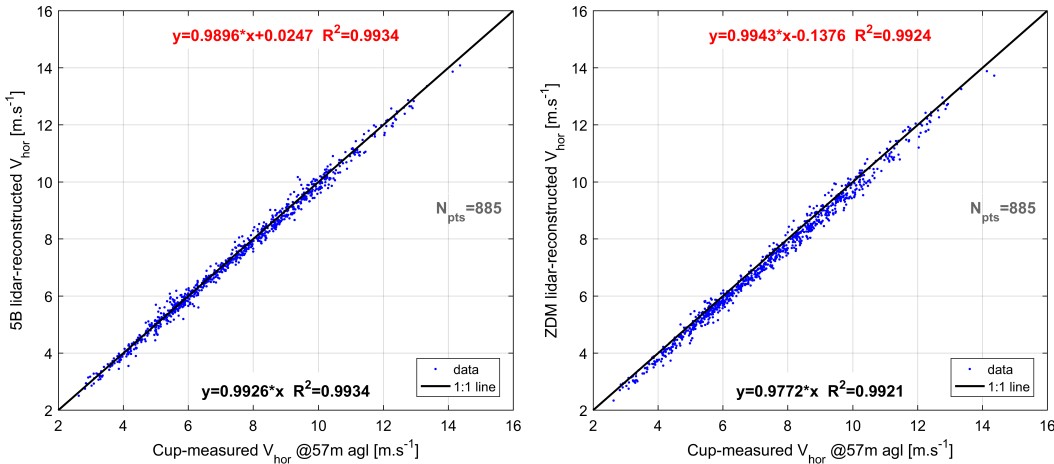

**Figure 10.** Comparison between mast-measured and lidar-estimated horizontal wind speed at $57.5\,\mathrm{m}$ height agl and $2.5D_{\mathrm{rot}}$ using short-range measurements. **Left**: 5B-Demo lidar, using 5 LOS and 4 ranges. **Right**: ZDM lidar, using 6 LOS and 3 ranges.



### 4.3 Summary of WFR results

A summary of all results is given in Table 2. Four cases of data filtering are analysed:

(1) Joint datasets for the restricted wind sector $[93°, 123°]$ (displayed in Fig. 7 and Fig. 9);

(2) Disjoint datasets for the restricted wind sector $[93°, 123°]$;

(3) Joint datasets for the 'IEC' undisturbed sector $[120°, 219°]$;

(4) Disjoint datasets for the 'IEC' undisturbed sector $[110°, 219°]$.

**Table 2.** Summary of comparison results between lidar-estimated and mast-measured horizontal wind speed, at hub height.

| Data filtering | | Reconstruction case | | Forced linear regressions results | | | |
|---|---|---|---|---|---|---|---|
| Case | Direction sector | Dataset | Lidar | Input measurement ranges | gain | $R^2$ | Number of periods |
| 1 | $[93°, 123°]$ | Joint | 5B-Demo, 5 LOS | 2.0 $D_{\mathrm{rot}}$ | 1.0146 | 0.9936 | 885 |
| | | | ZDM, 6 LOS | 2.5 $D_{\mathrm{rot}}$ | 1.0090 | 0.9938 | |
| | | | 5B-Demo, 5 LOS | from 0.5 to 1.15 $D_{\mathrm{rot}}$ | 1.0063 | 0.9944 | |
| | | | ZDM, 6 LOS | from 0.3 to 1.25 $D_{\mathrm{rot}}$ | 0.9961 | 0.9947 | |
| 2 | $[93°, 123°]$ | disjoint | 5B-Demo, 5 LOS | 2.0 $D_{\mathrm{rot}}$ | 1.0133 | 0.9953 | 1476 |
| | | | ZDM, 6 LOS | 2.5 $D_{\mathrm{rot}}$ | 1.0080 | 0.9942 | 2143 |
| | | | 5B-Demo, 5 LOS | from 0.5 to 1.15 $D_{\mathrm{rot}}$ | 1.0057 | 0.9961 | 1123 |
| | | | ZDM, 6 LOS | from 0.3 to 1.25 $D_{\mathrm{rot}}$ | 0.9965 | 0.9962 | 2659 |
| 3 | $[110°, 219°]$ (IEC free sector) | Joint | 5B-Demo, 5 LOS | 2.0 $D_{\mathrm{rot}}$ | 1.0059 | 0.9848 | 2815 |
| | | | ZDM, 6 LOS | 2.5 $D_{\mathrm{rot}}$ | 1.0028 | 0.9841 | |
| | | | 5B-Demo, 5 LOS | from 0.5 to 1.15 $D_{\mathrm{rot}}$ | 0.9997 | 0.9877 | |
| | | | ZDM, 6 LOS | from 0.3 to 1.25 $D_{\mathrm{rot}}$ | 0.9923 | 0.9885 | |
| 4 | $[110°, 219°]$ (IEC free sector) | disjoint | 5B-Demo, 5 LOS | 2.0 $D_{\mathrm{rot}}$ | 1.0041 | 0.9840 | 4588 |
| | | | ZDM, 6 LOS | 2.5 $D_{\mathrm{rot}}$ | 1.0038 | 0.9860 | 5615 |
| | | | 5B-Demo, 5 LOS | from 0.5 to 1.15 $D_{\mathrm{rot}}$ | 0.9988 | 0.9888 | 4099 |
| | | | ZDM, 6 LOS | from 0.3 to 1.25 $D_{\mathrm{rot}}$ | 0.9935 | 0.9897 | 6199 |

In the disjoint case, filters are applied independently to each lidar and reconstruction cases, and then combined with the mast and turbines filters. In the joint case, all filters are combined.

Cases (1) and (2) show an overestimation of $1 - 1.5\%$ between the lidar estimates and mast measurements using the wind

model and a single lidar measurement range. In the undisturbed sector (cases (3) and (4)), the overestimation is only of $0.5\%$, which may be attributed to the mast being most of the time outside the turbine's induction. Moreover, the coefficients of determination values drop significantly, to approximately $0.9850$. A plausible explanation is the larger separation between the lidar measurement points and the mast location causing decorrelation between the wind speed signals.

Multi-distance reconstruction cases including the simple induction model provide robust estimates of wind speed at hub

height, with observed gain errors within $0.5\%$. Retrieving accurate estimates of free stream wind characteristics based on





nacelle lidar near flow measurements thus proves to be feasible. However, the wind speed comparison results are not as consistent at the $57\,\mathrm{m}$ height agl. Using the short-range measurements, the covered range of heights is more narrow and the quality of fit of the shear characteristic may be impaired.

In all of the four cases, 5B-Demo overestimates the wind speed by $0.5\,\% - 1.0\,\%$ compared to ZDM. Comparisons in $V_{\mathrm{los}}$ between the two lidars were performed for closely located measurement points. The difference in reconstructed speed is consistent with the difference observed in $V_{\mathrm{los}}$ comparisons. Correcting the lidars $V_{\mathrm{los}}$ measurements according to the calibration relations would bring the speed estimates from both systems closer, but cannot fully explain the difference. In cases (2) and (4), the valid number of data points is lower for 5B-Demo than for ZDM. This is due to dust and salts accumulating on the 5B-Demo windows during Summer, causing lower power levels in the emitted and backscattered signals, and to the lack of an automatic cleaning system for this prototype lidar. The issue is more significant at ranges further from the 5B-Demo lidar's focus point: for example, the valid periods are twice as numerous when applying the wind model at the $1.0 D_{\mathrm{rot}}$ rather than the $2.0 D_{\mathrm{rot}}$ measurement range.

Plausible explanations for biases between the two WFR models are:

- signal extinction at long ranges: can yield lower quality $V_{\mathrm{los}}$ measurements;

- lower coherence at long ranges: due to increased spatial separation between $V_{\mathrm{los}}$ measurement locations;

- lidar volume averaging effects: at large distance, and for LOS oriented downwards, the lidars probe heights where strong non-linear wind shear occurs;

## 5   Discussions

### 5.1   On modelling improvements in lidar WFR

The induction model used in this paper is one-dimensional. It accounts only for the streamwise variation of the flow and neglects any radial dependency of the induction. The induction model therefore assumes constant loading of the rotor. In reality, the thrust generation varies with the radial coordinate due to the blades aerodynamic profile.

Within the UniTTe project, Reynolds-Average Navier-Stokes simulations were carried out for a variety of turbine sizes and rotor designs (Troldborg and Meyer Forsting, 2016). The turbine-induced flow field proves to be self-similar. A two-dimensional engineering model of the induction was also proposed, by adjusting Eq. 10. Figure 11 displays analytical induction flow fields respectively generated with the one- (left) or two-dimensional models (right). Although the radial evolution of the induction seems to be only significant at distances lower than two rotor radii, such a more advanced description of the flow field may be implemented as part of the WFR and improve the wind-induction model adequacy.

Regarding lidar modelling, we performed in this paper point-like simulations of $V_{\mathrm{los}}$ measurements. The lidar model could be enhanced by integrating the lidar probe volume weighting function. The simulation of the $V_{\mathrm{los}}$ measurement would then be carried out by choosing a discrete number of points along the lidar beam path and associating weights to each point.



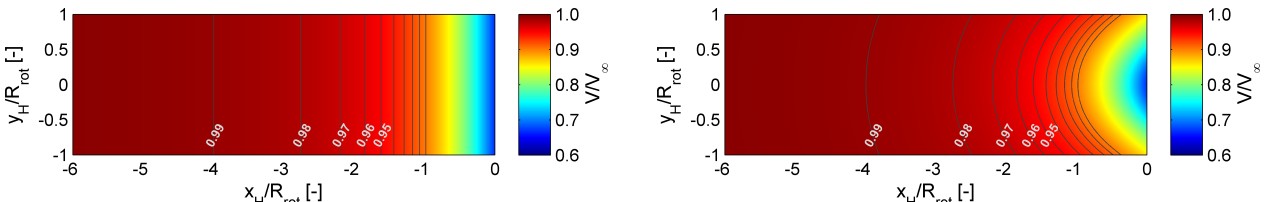

**Figure 11.** Analytical flow field in the induction zone of a wind turbine, at hub height and with an induction factor $a = 0.334$.
**Left:** one-dimensional model (Eq. 10). **Right:** two-dimensional model.

## 5.2 On free stream wind in power performance testing

In the power performance norms (IEC 61400-12-1 and IEC 61400-12-2), free stream wind speed is defined as *"the horizontal component of free stream wind that would exist at the position of the centre of the turbine's rotor if the turbine were not present"*. It is therefore impossible to measure free stream wind directly.

The wind speed measured by a cup anemometer top-mounted on a met. mast typically located $2.5D_{\mathrm{rot}}$ from the turbine is only an approximation of $V_\infty$. When the turbine is closely aligned with the mast and operates at high thrust coefficient (below rated speed), the $2.5D_{\mathrm{rot}}$ wind experiences already a deficit in speed of the order of $0.5\%$ or more. In opposition, the combined wind-induction model estimates the true free stream wind speed, the $V_\infty$ characteristic.

## 5.3 Advantages and limitations of measuring with a mast or nacelle-based lidars

The mast-based power performance procedures were originally designed for turbines of much smaller size than modern megawatt ones. For large modern turbines, their limitations are:

– the uneconomical cost of tall masts, particularly offshore;

– the decorrelation between power and wind speed signals: for a $120\,\mathrm{m}$ rotor diameter turbine, the mast must be placed $300 - 500\,\mathrm{m}$ from the turbine. Over such large distances, significant decorrelation phenomena occur;

– reduced undisturbed wind sector: e.g. when the mast is in turbines' wakes.

Mast-based measurements rely on well-established anemometry techniques (cup or sonic anemometers). This constitutes the main strength of current power performance procedures.

Contrary, WFR from lidars requires hypotheses on the flow inherent to their measurement principles. These hypotheses may be questionable. However, we demonstrated in this paper that model-fitting WFR from nacelle-based lidar short-range measurements takes advantage of the enhanced spatial information – the wind being probed at multiple heights and upstream distances – and provides robust estimates of true free stream wind.

Lidar short-range measurements techniques overcome both the current technological range limitation of nacelle-based systems and the aforementioned signal decorrelation issues. Additionally, close to the turbine, induction effects are anticipated to



prevail over terrain ones (Forsting et al., 2016). Short-range nacelle lidar measurements might also open the path towards free stream wind estimations in situations where it cannot be measured, such as in complex terrain or perhaps in an offshore array (due to wakes interaction).

## 6 Conclusion

We presented in this paper, a wind field reconstruction technique applicable to nacelle-based profiling lidars and providing wind speed estimations designed to be suitable for power performance verification. The method fits 10-minute averaged lidar measurements to an assumed wind model by minimising the error between lidar-measured and wind model-estimated line-of-sight velocities.

Experimental data from a 7-month measurement campaign conducted in Nørrekær Enge, Denmark, was used to compare wind field characteristics estimates obtained with nacelle lidars and an IEC-compliant meteorology mast. Identical wind field reconstruction algorithms were applied independently to two commercially developed nacelle lidars.

The profiling capabilities of the *5-beam Avent Demonstrator* and of the *ZephIR Dual-Mode* lidar systems allowed to define flow models yielding estimates of wind speed, direction and vertical shear. Such a wind model was applied to measurements taken first at a single distance far upstream the turbine. The model proved its ability to provide consistent wind speed estimations at several heights: lidar-estimated and mast-measured wind speeds agreed with an error of approximately $1-1.5\,\%$.

Next, the turbine's induction was accounted for by integrating a simple induction model – derived from the Vortex Sheet and the actuator disk theories – into the reconstruction algorithms. Utilizing the combined wind-induction model, free stream wind characteristics were estimated by fitting lidar measurements taken at several distances close to the rotor. This innovative method provides robust estimates of the free stream wind speed. Wind speeds reconstructed at the mast distance and hub height were within $0.5\,\%$ of cup anemometer measurements.

The developed reconstruction algorithm can be applied to any type of nacelle-based wind lidar system and any type of wind turbine rotor.




*Author contributions.* Antoine Borraccino conducted the research work and wrote the paper. David Schlipf and Florian Haizmann extensively contributed to the development of the wind field reconstruction method. Rozenn Wagner supervised the research work and contributed to the structure of the paper. All co-authors participated in the conception of the manuscript.

*Competing interests.* The authors declare that they have no conflict of interest.

5   *Acknowledgements.* The research work carried out and reported in this paper was performed under the Unified Turbine Testing (www.UniTTe.dk) project lead by *DTU Wind Energy* and funded by *Innovation Fund Denmark*. The two lidar systems providing the measurement data used in the study were kindly provided by the manufacturers, *ZephIR Lidar* and *Avent Lidar Technology*. The authors are thankful for the support, in particular to Matthieu Boquet and Paul Mazoyer on Avent's side, and to Michael Harris and Chris Slinger on ZephIR's side. Thanks to *Vattenfall* and *Siemens Wind Power* for providing the site and turbine to conduct the measurement campaign. Thanks also to Andrea Vignaroli,

10   Carsten Weber Kock and *DTU Wind Energy*'s Test and Measurement section whose work ensured the acquisition of high quality datasets, and to Rozenn Wagner for all the valuable advice.

Finally, special thanks to *Stuttgart Wind Energy*'s group for welcoming Antoine Borraccino during a 3-month stay. This work would not have been possible, and certainly not as productive, without the support and contributions of David Schlipf and Florian Haizmann.



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
