# Peer review of "Wind Field Reconstruction from Nacelle-Mounted Lidars Short Range Measurements"

_Wind Energy Science, 2017_

## Referee Comment (RC1) · D. Di Domenico (Referee) · 14 Apr 2017

The authors present the experimental results of a strategy of Wind Field Reconstruction from two LiDARs Nacelle-Mounted measurements by comparing the estimated wind speed to a classical mast sensors (cup anemometer) measurement. The two LiDARs, namely a 5-beam Avent and a ZephIR Dual-mode, are able to collect several wind speed measurements at different distances and directions from the wind turbine. These measurements are used to identify the parameters of a wind field model (that includes the induction), by using a Damped Least-Squares method, which aims at minimizing the error between the Lidar measurements and the model prediction. The model is used to reconstruct the wind speed in several points, including the mast sensors positions (2.5 times the rotor diameter from the WT). This allows to compare the estimation to the measurements and to evaluate the performance of the reconstruction algorithm. The paper is well written and the topic is relevant. The Lidar, the wind and the induction models are explained clearly and in details, explaining the meaning of the parameters to be identified (WFC) and their impact on the model. The results are well detailed and discussed. Minor remark: The minimization problem and the optimization algorithm are just briefly described, in particular if compared to the model description. More details could be added or discussed in details, as the convexity of the problem and the impact of the algorithm tuning parameters (damping parameter or tolerance, for instance) on the algorithm results.

———————————————————

---

## Referee Comment (RC2) · S. Feeney (Referee) · 19 Apr 2017

I found this paper to be very well written and both a comprehensive review of the subject and a significant result given the quality of the agreements between the measuremed and modelled Wind Field Characteristics. It gives me hope that techniques can be developed to allow a meaningful performance test for wind turbines in complex terrain and for the next generation offshore turbines. Well done to all.

I have a few minor comments that are listed below which the author may consider reviewing:

Page 1 line 2: I don't like the structure of the second sentence. It makes it seem like

[Figure]

the industry has been battling with this issue for many years whereas in reality profilling nacelle LiDARs are relatively new.

Page 2 Line 1: Could a source be added for the 0.7% figure? In my experience 1% is what is assumed in the industry (and the IEC) so I would be interested to see where the 0.7% comes from. Later in the paper the figure 0.5% is used which I understand comes from the results presented in the paper? This figure is important to AEP calculations so it should be as clear as possible where the numbers are coming from

Page 2 Line 31: I don't agree that the vertical shear profiles are arbitrary (definition: "based on random choice or personal whim, rather than any reason or system")

Page 9 Line 18: The sentence describing the site is poorly worded. Remove the word "the" before terrain flattness?

Page 11 Line 3: Replace We with the authors or an algorithm was developed

Page 15 Line 12 -14: I don't understand why the 10m range was discarded and the link to self-similarity. This seems like it is contradicting the discussion on self-similarity earlier in the paper? Please consider rewording this section

Page 16 line 6: Could the relative lack of agreement also be caused by mast effects on the side boom mounted anemometer?

---

## Author Response (AR1)

Dear Dr. Di Domenico,

Thank you for your time and comments on our article. We appreciate the positive feedback.

Your minor remark on the details of the minimisation problem and optimisation algorithm is helpful. The points you raised had been considered during the development of our Wind Field Reconstruction methods. Here are our comments:

\_\_\_\_\_\_\_\_\_\_\_\_\_\_\_\_\_\_\_\_\_\_\_\_\_

[Figure]

**1) on the problem convexity**

The convexity of the "cost function" of the minimisation problem would ensure the uniqueness of the optimal solution, i.e. the found local minimum is also a global minimum.

To test the convexity of a multi-dimensional function, one would formally need to derive or numerically approximate the Hessian matrix of the cost function and determine whether it is (semi)-definite positive. We chose a more basic approach where we tested the variability of the fitted WFC vector solution with a number of different initial values (thus allowing to start the optimisation from a farther point in the multi-dimensional space). Even with rather silly initial values (e.g. speed of -300m/s, yaw misalignment of 180deg, negative induction factor, etc), the results were found to be identical.
* * *
**2) on the algorithm tuning parameters**

We used the default damping parameter of the MatLab-integrated Levenberg-Marquardt algorithm. Changing the damping parameter value would mainly result in a change of convergence speed, zhich has never been a problem in this study.

The tolerance values – both on the V_los residual vector and on the change in WFC vector over the last optimisation iteration – were tested. The fitted WFC values proved to be negligibly sensitive to the tolerance values, for all values smaller than the default ones. Therefore the default values were selected for the results presented in the paper.
* * *
In the submitted version of the article, we chose to avoid such highly technical details in order to focus on the most central parts of the model-fitting WFR. However, it appears valuable to mention those points. We will most probably add a couple paragraphs in Section 2.2 ("Formulation and solving of the minimisation problem"), based on this interactive discussion.

[Figure]

[Figure]

Wind Energ. Sci. Discuss.,
doi:10.5194/wes-2017-10-AC2, 2017

[Figure]

We thank you for your time and comments on the article. Your positive feedback is much appreciated. Below is our reply to your comments before the revision of the article, point by point.

——————— Page 1, line 2:

It is correct that such lidar systems are rather recent.

Revision: the sentence will be reformulated.

[Figure]

————————— Page 2, line 1:

the 0.7% comes from applying the simple induction model (eq. 10) with a canonical value of axial induction factor equals to 1/3. This 0.7% value is conservative for an isolated turbine – as the turbine-mast direction is rarely perfectly aligned with the wind direction. It is NOT conservative in the case of a row of turbines or in the case of a complete wind farm (e.g. offshore): wind farm effect extends the induction zone further than in the case of a single turbine, thus yielding higher axial induction factors.

The 0.5% value used further in the paper is indeed coming from the presented results (see page 17, lines 10-11).

Revision: adding details on the source.

————————— Page 2, line 31:

These paragraphs were discussed with the two lidar manufacturers, as they concern the in-house reconstruction algorithms.

Revision: removing "arbitrary"

————————— Page 9, line 18:

Revision: removing "the".

————————— Page 11, line 3:

The point here is to inform that this algorithm may differ slightly from the one of the lidar manufacturer.

Revision: suggestion accepted.

————————— Page 15, lines 12-14:

Unfortunately, the article we refer to is not yet published. The work reported in this article shows that the self-similarity assumption (essential for using the same simple induction model for any wind turbine) is not valid very close to the turbine rotor. This is

because the blades' aerodynamic profiles will differ from one rotor design to another, and that the aerodynamic profiles was shown to matter only very close to the rotor plane. Hence the 10m range was discarded from the analysis.

Moreover, this range was poorly sampled with the used lidar configuration: its purpose was only to provide a means to detect fog events.

Revision: none planned.

——————— Page 16, line 6:

On principle, it could be due to some extent to mast shadowing effects. The boom direction was however checked and the cup anemometer's measurement should not be affected that significantly.

Revision: maybe add a comment on this.

———————————————

[revised manuscript text omitted]
 $2.3\,\text{MW}$ turbines having a rotor diameter $D_{\text{rot}}$ of $93\,\text{m}$ and a hub height $H_{\text{hub}}$ of $80\,\text{m}$ agl. The orientation of the
* * *
[1]Find more information on: https://corporate.vattenfall.dk/

turbines' row is $75° - 255°$. The site is mainly characterised by open crop fields and flat terrain. In the vicinity of turbine number 4 (T04), except for the turbines' foundations, variations in elevation of $\pm 1\,\mathrm{m}$ are observed (see Fig. 3). The prevailing wind direction is West. In Jutland, such western winds often feature high speeds (Peña et al., 2016).

[Figure]

**Figure 3. Left:** elevation map in the vicinity of T04, Nørrekær Enge wind farm. Map: DHM/Terrain (0.4 m grid). Source: *Styrelsen for Dataforsyning og Effektivisering*. **Right:** wind rose during the NKE campaign, measured by the top-mounted cup anemometer and wind vane.

**3.2 Meteorological mast and turbine instrumentation**

5   A meteorological mast was installed $232\,\mathrm{m}$ from T04 approximately in the $103°$ direction (see Fig. 3). The mast instrumentation complies with the requirements of the standards for power performance measurement (IEC61400-12-1, 2005):

– one top-mounted cup anemometer at $80\,\mathrm{m}$ agl.;

– three cup anemometers and wind vanes at $33.5\,\mathrm{m}$, $57.5\,\mathrm{m}$ and $78\,\mathrm{m}$ agl.;

– one sonic anemometer at $76\,\mathrm{m}$ agl.;

10   – other sensors: air temperature at $2\,\mathrm{m}$ and $78\,\mathrm{m}$, relative humidity at $78\,\mathrm{m}$, atmospheric pressure at $77\,\mathrm{m}$, and precipitation at $20\,\mathrm{
[revised manuscript text omitted]